# Refining Evaluation of Bone Mass and Adipose Distribution in Dunnigan Syndrome

**DOI:** 10.3390/ijms241713118

**Published:** 2023-08-23

**Authors:** Mariana Lima Mascarenhas Moreira, Iana Mizumukai de Araújo, Sandra Yasuyo Fukada, Lucas Gabriel R. Venturini, Natalia Rossin Guidorizzi, Carlos Ernesto Garrido, Clifford J. Rosen, Francisco José Albuquerque de Paula

**Affiliations:** 1Department of Internal Medicine, Ribeirão Preto Medical School, University of Sao Paulo, Ribeirao Preto 14049-900, SP, Brazil; mahmasca@yahoo.com.br (M.L.M.M.); iana.mizumukai@gmail.com (I.M.d.A.); naguidorizzi@gmail.com (N.R.G.); 2Department of Biomolecular Sciences, School of Pharmaceutical Sciences of Ribeirao Preto, University of Sao Paulo, Ribeirao Preto 14040-900, SP, Brazil; sfukada@usp.br (S.Y.F.); lucasventurini@usp.br (L.G.R.V.); 3Department of Physics, Faculty of Philosophy, Sciences and Letters of Ribeirao Preto, University of Sao Paulo, Ribeirao Preto 14049-900, SP, Brazil; garrido@ffclrp.usp.br; 4Maine Health Institute for Research, Scarborough, ME 04074, USA; cjrofen@gmail.com

**Keywords:** insulin resistance, bone mineral density, lipodystrophy, osteocalcin, TBS

## Abstract

Familial partial lipodystrophies (FPLD) are rare diseases characterized by selective loss of subcutaneous adipose tissue at different sites. This cross-sectional observational study aimed to estimate adipose tissue in the bone marrow (BMAT), intra (IMCL) and extra-myocyte lipids (EMCL), and define the bone phenotype in the context of FPLD2/Dunnigan syndrome (DS). The subjects comprised 23 controls (C) and 18 DS patients, matched by age, weight and height. Blood samples, dual-energy X-ray absorptiometry for bone mineral density (BMD) and trabecular bone score (TBS) and 1H-spectroscopy using magnetic resonance to estimate BMAT in the lumbar spine, IMCL, EMCL and osteoclastogenesis were assessed. The prevalence of diabetes mellitus was 78% in DS patients. Glucose, HbA1c, triglycerides, insulin and HOMA-IR levels were elevated in DS, whereas HDLc, 25(OH)D, PTH and osteocalcin levels were reduced. BMD was similar between groups at all sites, except 1/3 radius, which was lower in DS group. TBS was reduced in DS. DS presented increased osteoclastogenesis and elevated BMAT, with greater saturation levels and higher IMCL than the C group. HOMA-IR and EMCL were negatively associated with TBS; osteocalcin and EMCL were correlated negatively with BMD. This study contributes to refining the estimation of adipose tissue in DS by showing increased adiposity in the lumbar spine and muscle tissue. DXA detected lower TBS and BMD in the 1/3 radius, suggesting impairment in bone quality and that bone mass is mainly affected in the cortical bone.

## 1. Introduction

Familial partial lipodystrophy (FPLD) is an umbrella term for a group of rare inherited disorders, characterized by regional loss of adipose tissue, in which Dunnigan syndrome (DS, FPLD2) is the most common form. DS has an autosomal dominant pattern of heritability, determined by a pathogenic variant of the laminin A/C gene (*LMNA*).

DS is a natural model of metabolic syndrome due to the genetic prioritization of white adipose tissue (WAT) development in the area of visceral adipose tissue (VAT), whereas the gluteofemoral region is almost devoid of WAT [1]. Typically, the subcutaneous adipose tissue (SAT) exists in the neck, face, axillary, and pubic regions. The metabolic environment of severe insulin resistance (IR) expedites the early emergence of type 2 diabetes mellitus (T2D) and steatohepatitis, and life expectancy is often shortened by cardiovascular diseases. Dysfunctional adipose tissue is a hub for the attraction of proinflammatory macrophages, the overspill of lipids into other tissues, and the generation of adipokines. These circumstances place diabetes mellitus and cardiovascular diseases as the main targets of most studies in the literature on DS, whereas other diseases have received scarce attention. In particular, few studies have evaluated osteoporosis and the influence of adipose tissue and muscle on bone mass in DS individuals. 

Currently, there is only one study that investigated bone mineral density in individuals diagnosed with DS. Fernández-Pombo and colleagues [2] reported that these patients have normal bone mass. However, the influence of IR on bone mass was not evaluated in that study. Most individuals diagnosed with obesity and T2D show normal or increased bone mass but are paradoxically not protected from fracture [3,4]. There are currently no data available in the literature on osteoporosis or fracture risk in patients with FPLD. 

In a recent study, it was verified that parameters strongly linked to insulin resistance (e.g., VAT, intrahepatic lipids and homeostasis model assessment—HOMA-IR) do not have a negative relationship with bone mass [5]. On the other hand, a negative association was observed between IR parameters and the trabecular bone score (TBS), an estimate of bone quality based on pixel gray-level variations in the lumbar spine dual-energy X-ray absorptiometry (DXA) exam [5,6]. Characteristically, obesity and T2D show reduced indices of bone remodeling, and in vitro osteoclastogenesis could possibly contribute to the knowledge in this area. Moreover, it was observed that bone mass is negatively associated with serum levels of osteocalcin in T2D. Although approximately 28–51% of the subjects diagnosed with DS show T2D [7], there are no studies on the biochemical markers for bone remodeling and TBS in subjects with DS. 

Observational studies show that obesity is associated with low serum levels of 25-hydroxyvitamin D—25(OH)D [8]. The exact mechanism involved in this endocrine disorder is still to be elucidated, but it is hypothesized that it may be due to the deposition of vitamin D within adipose tissue. The present study evaluated circulatory levels of PTH and 25(OH)D in individuals diagnosed with DS.

There is little data in the literature on the accumulation of lipids in extra (EMCL) and intra-myocyte (IMCL) spaces and in bone marrow adipose tissue, as well as the amount of saturated and unsaturated lipids within the bone marrow in DS. A previous study that evaluated only three individuals suggested preservation of intermuscular and bone marrow adipose tissue in DS [9], while a more recent evaluation of IMCL in FPLD patients showed higher content compared to controls and athletes and positive associations with HOMA-IR, whereas EMCL exhibited no significant difference between groups [10]. Ectopic fat may be deposited in layers along the muscle (EMCL) or as fat droplets within the myocyte (IMCL) [11]. It is well known that insulin resistance strongly correlates with IMCL [12], and the IMCL/EMCL ratio is also accepted as a parameter linked to insulin resistance [13], but the role of EMCL is still debated. 

Bone marrow adipose tissue (BMAT) appears during perinatal life and is programmed to expand thereafter. There are studies showing no difference in the amount of BMAT between T2D and obese individuals compared to controls [9]. In line with these results, Patsch and colleagues observed a normal amount of BMAT in T2D, suggesting that BMAT is not a niche for the storage of lipids in conditions of energy surplus [14]. In addition, they verified that T2D and fractures are associated with saturated lipids within the BMAT, indicating that the quality of lipids is a factor that may influence bone resistance [15,16]. Usually, although not always, BMAT expansion and bone loss are concomitant occurrences, as osteoblasts and marrow adipocytes are alternative routes of differentiation of the same mesenchymal stem cell. However, the relationship between BMAT and bone mass in humans and mice with congenital lipodystrophy may be dependent on each specific disease mutation. 

This study takes advantage of the versatility of nuclear magnetic resonance to refine the estimation of adipose tissue distribution in individuals diagnosed with DS, quantitatively verifying BMAT and its fractions, muscle adipose tissue (EMCL) and myocyte lipids (IMCL). Also, the present study was designed to contribute further to the characterization of the bone phenotype in DS patients, including the assessment of osteoclastogenesis, and to evaluate the relationship between bone mass and parameters of insulin resistance in DS.

## 2. Results

### 2.1. Clinical and Laboratory Findings

Table 1 shows the clinical and laboratory characteristics of the two groups evaluated in this study. There were no differences in age, body weight, and height between groups. All individuals in the DS group had the clinical phenotype of FPLD (apparent muscle hypertrophy in members, phlebomegaly and expansion of fat mass in the neck, hump and genital areas). Of these, 12/18 (67%) had genotype characterization of the *LMNA* mutation. As expected, the serum levels of leptin were reduced in the DS group, albeit not significantly (*p* = 0.07), and no difference in chemerin levels was observed between the groups. 

Table 1 shows the negative impact of DS on energy metabolism, with elevated levels of glucose, HbA1c, insulin and HOMA-IR. We observed a very high prevalence of diabetes mellitus (DM) (14/18; 78%) in DS subjects, as well as early onset of DM, with a mean diagnosis duration of 11.7 ± 5.9 years. In this subset of DM patients, 8/14 (57%) were already insulin users, 11/14 (79%) used metformin, the most prevalent antidiabetic drug in the cohort, and 3/14 (21%) used pioglitazone. 

Only three patients, two of whom were from the same family, were using pioglitazone. The small number of individuals using pioglitazone was due to the restricted access to this agent, as it is not included in the list of drugs distributed by the government’s healthcare program. Only one of the three patients used pioglitazone regularly for more than one year, whereas the others used it for a few months. Nevertheless, only one individual from this subgroup had low BMD for their age (LS Z-score −2.1SD; TH Z-score −1.8SD; FN Z-score −1.9SD); the others had normal bone mass. These three patients also had lower body weight (mean = 55.3 kg) and BMI (mean = 22.1) compared to DS. The impact of pioglitazone on FPLD2 bone mass needs further evaluation. None of the patients in the C group used antidiabetic agents. 

The biochemical alterations were not limited to glucose metabolism, they also showed severe dyslipidemia. DS group had lower HDLc and higher serum levels of triglycerides and hepatic enzymes than the C group (Table 1) in spite of 67% of the patients using statins (12/18) and 39% (7/18) using fibrates. 

DS group showed striking differences in mineral and bone metabolism. DS patients showed significantly reduced levels of 25(OH)D. PTH serum levels were also lower in DS compared to C, but no difference was observed in calcium or phosphate concentrations between the groups. Only 2/18 (11%) DS subjects used vitamin D tablets and 1 patient supplemented calcium. The levels of physical fitness were not assessed in the C or DS group. In addition, osteocalcin levels were significantly reduced in DS compared with the C group. 

### 2.2. Bone Phenotype, Adipose Tissue Distribution and Osteoclastogenesis

The BMD and respective Z-scores are shown in Table 2. 

There were no differences in BMD and Z-score between groups in the lumbar spine, femoral neck or total hip, except for 1/3 radius, which was lower in the DS group. Likewise, TBS values were significantly lower in the DS than in the C group (*p* < 0.01) (Figure 1A). All individuals from the control group were classified as low risk of fracture (23/23 with TBS > 1.310), based on the TBS estimation [17]. On the other hand, in the DS group, 5 subjects (28%) had an intermediate risk and 4 subjects (22%) had a high risk of fracture while the remaining 9 (50%) showed a low risk of fracture, based on the evaluation of TBS (Figure 1B).

Figure 2A shows that IMCL (C = 700.6 ± 413.2 vs. DS = 1244.9 ± 777.4 UA; *p* < 0.05) increased in the DS group. EMCL (C = 2439.9 ± 1107.9 vs. DS = 3461.2 ± 2427.0) and IMCL/EMCL ratio (C = 32.4 ± 17.7 vs. DS = 46.3 ± 28.4) were greater in the DS group, though not significantly. In addition, the DS group showed a higher amount of BMAT than C (C = 32.7 ± 9.6% vs. DS = 50.0 ± 17.8%; *p* < 0.01), as shown in Figure 2B, and the qualitative analysis of lipids showed that the saturated fraction was also elevated in the DS group (31.8 ± 11.2%) compared to the C group (23.2 ± 8.9%), *p* < 0.05. There were no differences in the unsaturated fractions between the groups. Lipids curves obtained from ¹H-spectroscopy for the L3 vertebra and soleus muscle are depicted in Figure 2C,D.

The evaluation of osteoclast differentiation (Figure 3) from peripheral blood mononuclear cells stimulated with receptor activator of nuclear factor kappa-B ligand (RANKL) showed an increased area (mm²) in the DS group compared to the C group (C = 0.95 ± 0.7 vs. DS = 1.85 ± 1.0 mm²; *p* = 0.05).

The Osteoclast area in the DS group stimulated with exogenous chemerin was significantly increased compared to the C group area (C = 0.76 ± 0.5 vs. DS = 1.92 ± 1.3 mm²; *p* < 0.05) and the osteoclast number was also increased in all mediums in the DS group, although not significantly. 

There was a positive correlation between the osteoclast area with the lumbar spine and the total hip BMD, which probably reflects the inflammatory systemic status of this patient. No statistically significant correlation was observed between the osteoclast number and the area with remaining BMD sites, BMAT or TBS, as described in Table 3.

### 2.3. Influence of IR Markers on Bone Quantity, Quality and Lipid Distribution

The global analysis of correlations between bone, BMAT, and biochemical parameters is displayed in Figure 4. It shows that HOMA-IR strongly correlated with IMCL and HbA1c, which are markers of insulin resistance.

The lumbar spine, femoral neck and total hip BMDs, and parameters of insulin resistance, such as HbA1c, HOMA-IR and IMCL, were not negatively associated. A 1/3 radius, on the other hand, showed a negative correlation with fasting glucose and HbA1c, maintained after non-adjusted linear regression. 

Osteocalcin had a negative relationship with BMD, confirmed in the LS region using regression analysis after adjustment for T2D, sex, age and BMI (β = −1.04; CI95 −1.99, −0.09; *p* < 0.05). Vitamin D and PTH showed no association with DXA parameters. Table 4 summarizes the main results of the linear regression analysis before and after adjustments (Appendix A).

TBS showed a negative correlation with fasting glucose and HbA1c, confirmed using linear regression, but not maintained after adjustments. Osteocalcin had no significant association with TBS. HOMA-IR and EMCL correlated negatively with TBS, and this effect persisted after adjusted regression (Figure 5A,B), while IMCL had a negative effect on TBS, but it was not significant in this cohort. A tendency towards a negative association between TBS and BMAT was observed in the regression analysis (*p* = 0.05), but this relationship was not maintained after adjustments (Figure 5C).

Besides having a negative association with TBS, EMCL also had a negative association with BMD, which persisted after adjusted regression analysis in the lumbar spine and total hip. The IMCL, IMCL/EMCL ratio, and BMAT and its fractions were not associated with BMD or TBS.

## 3. Discussion

In a contemporaneous scenario in which metabolic syndrome is constantly rising, FPLD is a rare but important condition that can help delineate the importance of adipose tissue to skeletal health, especially considering the impacts of ectopic lipid deposits and lipotoxicity. This condition also provides a unique clinical model to evaluate the impact of severe insulin resistance on the bone. The present study observed that DS manifests precocious diabetes, dyslipidemia, and nonalcoholic metabolic disorder in the liver. BMD is preserved in the lumbar spine and proximal femur; however, the 1/3 radius, a site representative of cortical bone, is significantly decreased. Nevertheless, the TBS captures architectural deteriorations in the lumbar spine of DS subjects. Remarkably, BMAT from the lumbar spine is not only preserved but increased in DS, indicating that this niche can potentially help to differentiate the several types of FPLD. 

A recent study reported that the prevalence of diabetes mellitus in Brazil was 7.7% in 2019, and approximately 72% were aged 55 years or older [17]. In the present DS cohort, 78% had diabetes mellitus; nonetheless, their mean age was only 42.6 ± 8.4 years and the patients exhibited significantly elevated levels of glucose and HbA1c compared to the C group. These results highlight the negative relationship between lipodystrophy of subcutaneous adipose tissue and energy metabolism, leading to the early onset of hyperglycemia, and severe deterioration of β-cell function. The clinical and laboratory clues of insulin resistance are acanthosis nigricans, arterial hypertension, decreased HDLc and elevated serum levels of triglycerides, ALT and AST, all of which were manifested in the DS cohort. The leptin values are related to fat mass depots and vary according to fat quantities. In the control group, the percentual body fat ranged from 22 to 52%, and in the FPLD group, it ranged from 15 to 36%. This could justify the high variation in the serum levels of leptin in both groups. In addition, leptin concentrations in FPLD patients are highly variable, and our findings are in line with the data published previously by Hegele et al., who described a leptin value of 11.1 ± 10.3 ng/mL in FPLD family members, and Brown et al., who described values of 5.9 ± 3.0 in FPLD patients [19,20].

In spite of the severe metabolic disturbance, the DS and control groups showed non-significant differences in the BMD values of the lumbar spine, femoral neck and total hip. On the other hand, BMD from the 1/3 radius was significantly reduced in DS individuals. Even though the Z-score was normal in both groups, the difference between them was 0.080 g/cm² (9.8%), which was higher than the 1/3 radius of the least significant change (LSC = 0.033) in the DXA equipment. It is necessary to consider that the samples comprised young adults, indicating that the lower BMD may impact fracture risk in the future.

There is a consensus in the literature that body weight positively correlates with BMD; however, there is controversy about the relative importance of muscle and adipose tissue in the determination of bone mass. There are studies suggesting that body fat is the main predictor of BMD at all bone sites [21]. On the other hand, more recent studies suggest a negative relationship between fat mass index and BMD and, at the same time, a positive association between bone and muscle mass [22]. 

The present findings in DS and previous investigations on generalized lipodystrophy indicate that partial or total shrinkage of adipose tissues does not lead to low bone mass at the sites usually used for the diagnosis of osteoporosis [23]. In addition, genetically modified mice recapitulating generalized lipodystrophy show increased bone mass throughout the body, which declines after adipose tissue transplantation [24]. These results indirectly support that muscle tissue has major participation in bone mass maintenance. In a previous study on congenital generalized lipodystrophy, sites with a high content of trabecular bone, such as the lumbar spine and ultradistal radius, had high BMDs and Z-scores. On the other hand, the site with the lowest BMD and Z-score was the 1/3 radius. However, in this study, there was no control group for comparison [25]. Moreover, 50% of the DS subjects had low TBS (TBS ≤ 1.310), a parameter recognized as an independent risk factor for fracture. A multicenter study is necessary to investigate the capability of TBS to predict fracture risk in DS individuals as occurs in type 2 diabetes mellitus [26].

Spuler and colleagues reported that muscle hypertrophy is part of the alterations in the profile of body composition in DS patients [27]. In addition, they found reduced expression of myostatin in muscle biopsies. In spite of this, the DS individuals showed higher fatigue during exercise. The present study shows that DS subjects display increased IMCL, a metabolic alteration linked to insulin resistance in the muscle [28]. This finding reveals that the increased concentration of triglycerides inside myocytes might be involved in an array of metabolic alterations triggered by increased free fatty acids that lead to insulin resistance, reduced glucose disposal and dysfunctional mitochondria [29]. EMCL, a site for adipose tissue hypertrophy in the muscle, was negatively associated with BMD at all sites, except for the 1/3 radius. While the fat accumulation in the muscle is directly associated with metabolic disorders, there was no negative relationship between the IMCL/EMCL ratio and BMD values. These results are in line with other studies that evaluated the association of BMD with parameters strongly linked to insulin resistance (e.g., HOMA-IR, VAT and intrahepatic lipids) [5,30]. Moreover, in agreement with previous studies, the current findings show that TBS has negative associations with the parameters associated with insulin resistance, such as HOMA-IR. 

The peripheral distribution of fat in DS individuals has a similarity with Cushing’s disease. The present study reveals that they also share the same profile concerning BMAT expansion [31]. Patsch and colleagues observed that the components of saturated lipids are especially increased in diabetics with fractures [14]. The DS group exhibits a higher bone marrow saturation than the control group. We can speculate that BMAT expansion is not affected by *LMNA* mutations in the same way as WAT and that it responds distinctly from WAT when regulated by stimuli such as caloric intake, fasting and insulin levels. In addition, further studies are necessary to evaluate the association of fat quality in the bone marrow with bone fractures in Dunnigan syndrome, as this cohort was relatively young and fracture occurrence was not assessed. 

In addition, this study shows that this complex and toxic metabolic environment has a significant negative relationship with mineral metabolism, bone mass quality, and BMAT. Despite the decreased amount of adipose tissue, DS individuals show low serum levels of 25(OH)D but normal calcium and phosphorus levels. The levels of vitamin D may not represent an actual state of deficiency in these patients; there may be an alteration in the sensitivity of the calcium sensor receptor. Moreover, systemic and chronic diseases generally present with low levels of vitamin D, but we have no explanation for the finding of non-elevated PTH in this scenario. Perhaps the decreased fat depots alter the metabolism and bioavailability of vitamin D and its metabolites. This subject deserves more study. The present study challenges the hypothesis that hypovitaminosis D status in obesity is secondary to adipose tissue sequestration, though more studies are needed. 

As usually observed under systemic conditions, 25(OH)D was below 20 ng/mL in DS [32]. In a cohort of individuals with congenital generalized lipodystrophy, vitamin D levels were above 20 ng/mL, and in partial lipodystrophy, data are scarce [25]. The negative association found between obesity and vitamin D is supposed to be secondary to the deposition of vitamin D within the adipose tissue. However, this seems less likely to be a mechanism due to lipodystrophic adipose depots, since DS patients have low body fat and are expected to have sufficient vitamin D. All these individuals live in a rural area in the northeast of São Paulo state in Brazil, a region with generous sunlight throughout most of the year. Moreover, the serum levels of PTH in the DS group were lower than those in the C group, suggesting that bone turnover is not driven by low 25(OH)D.

Finally, an increased potential for osteoclastogenesis in peripheral blood monocytes was observed. The adipokine chemerin is expressed in WAT, and studies on humans revealed a positive relationship between chemerin concentrations and obesity, BMI, waist-to-hip ratio, abdominal circumference and VAT [33,34]. Previous studies by Murugunadan and colleagues [35,36] showed chemerin action in mesenchymal stem cells of BMAT, favoring adipose tissue expansion at this site. Additionally, they observed a positive influence on hematopoietic cell differentiation into osteoclasts and bone resorption, which was reversed after the use of chemerin receptor antagonists [37]. The inhibition of chemerin and its receptor also showed a positive association with thermogenic beige fat in animals and in vitro models [38]. In the present study, DS patients exhibited elevated BMAT and increased osteoclast area compared to controls when stimulated with chemerin in vitro; however, we did not observe elevated levels of chemerin in individuals with DS. These results most likely reflect a chronic inflammatory status associated with insulin resistance driven by lipodystrophy or a primary resorption abnormality in DS, rather than a direct influence of chemerin. A positive correlation between osteoclast area and lumbar spine/total hip BMD, which probably reflects the inflammatory systemic status of the patients, was observed. Most likely, the peripheral environment has a positive influence on the differentiation of monocytes into osteoclasts in vitro. On the other hand, the results obtained in peripheral monocyte culture do not reflect osteoclastogenesis in the bone microenvironment, as the circulatory levels of osteocalcin suggest a low bone turnover.

This study has some limitations. It has a cross-sectional design, and we could only evaluate the associations between the variables studied. Also, DS is a rare condition, which is an important obstacle to obtaining larger sample sizes and drawing more robust evidence. In spite of this, the cohort has a considerable number of individuals diagnosed with DS, in alignment with other studies, and is the first to include BMD, TBS, and adipose tissue evaluation in BMAT and muscle. In addition, MRI used to estimate lipids at the above-mentioned sites is considered the gold standard for this analysis, which is essential for the acquisition of original results on the distribution of adipose tissue and the association between metabolic and bone disorders. 

In summary, DS is a severe metabolic disease that exhibits a precocious incidence of diabetes mellitus and rapid deterioration in beta cell function. Nonetheless, the chaos observed in classical insulin-sensitive tissues seems to be less aggressive to the bone. BMD is preserved in the lumbar spine, femoral neck, and total hip, but is diminished in the cortical-rich 1/3 radius. TBS also captures textural alterations in the lumbar spine of DS subjects. There is no negative relationship between bone mass and the parameters of insulin resistance in DS patients. However, regarding bone quality, we observed that TBS is negatively associated with insulin resistance in DS. Furthermore, the present study contributes to the development of the phenotype of adipose tissue distribution in DS, since these individuals show increased BMAT, with more saturated fat, and increased intramuscular adipose tissue. This study encourages further investigation of fracture risk in DS patients.

## 4. Materials and Methods

### 4.1. Study Design and Population

This was a single-center cross-sectional observational study, in which 18 patients with DS (15F; 3M) were compared to 23 controls (18F; 5M) matched by sex, age, weight and body mass index (BMI). Since DS is a rare disease, with approximately 500 reported cases in the literature [39,40,41,42], and FPLD’s estimated prevalence is 2.8 cases/million [43], this study used a convenience sample of FPLD patients who already attended our diabetes outpatient clinics. Healthy controls were recruited from local areas using advertisements. Clinical data were obtained from the medical records of subjects; fertility, menstrual cycle regularity, and polycystic ovary syndrome were not evaluated in the study. Controls and patients’ demographic data (age, height and weight) and clinical information (history of diseases and medication use) were obtained from direct interviews. 

Among the 18 DS patients, 9 carried typical p.Arg482Trp *LMNA* mutation, 2 carried p.Arg644Cys *LMNA* mutation, 1 carried p.Ser583Leu *LMNA* mutation, 1 carried p.Gly215Glu *LPL* mutation and, in 5 subjects, mutations were yet to be identified and the diagnosis of FPLD was based on family history, physical examination, laboratory and imaging evaluations. Within the *LMNA* 482-codon mutation group, 4 belonged to the same family, and the other 5 were unrelated. The 2 patients with *LMNA* 644-codon mutation were siblings. The individual with a mutation in LPL had a typical clinical phenotype and a new genetic analysis was programmed. Among the 5 subjects without identified mutations, one was related to the siblings with p.Arg644Cys *LMNA* mutation and the other 4 had all clinical features, but no personal or familial molecular diagnosis of FPLD yet. 

The patients without confirmed mutation were included because they exhibited a clinical and biochemical profile found in FPLD, i.e., metabolic syndrome, presence of DM (glucose mean = 179.8 vs. 175.4 mg/dL), high triglycerides (mean = 477.3 vs. 453.7 mg/dL)/low HDLc (mean = 34 vs. 33 mg/dL) and altered body composition with elevated fat mass ratio index (mean = 1.58 vs. 1.56); BMD also was comparable (LS 1.125 vs. 1.180; TH 1.110 vs. 1.040; FN 1.080 vs. 1.020; 1/3 radius 0.900 vs. 0.810 g/cm²). According to Endocrine Society guidelines, genetic confirmation is not an obligatory criterion for diagnosis of Dunnigan Syndrome. Confirmatory genetic testing is helpful, but not required for the diagnosis of FPLD, since there is building evidence that additional loci for genetic lipodystrophies exist but are yet to be identified. Thus, negative genotyping is not sufficient to rule out a genetic condition when clinically suggestive [44].

The control group was composed of 23 non-lipodystrophic patients. The exclusion criteria were as follows: age below 18 yo or above 60 yo; the presence of a chronic disease known to affect bone metabolism; severe renal and/or liver disease; abnormal thyroid functioning; glucocorticoid, anticonvulsant or osteoporosis therapy (bisphosphonates, denosumab, teriparatide and calcitonin).

### 4.2. Biochemistry

Blood samples were collected after a 12h overnight fast. Biochemical measurements of calcium, glucose, glycated hemoglobin (HbA1c), total cholesterol and fractions (HDLc, LDLc), triglycerides, phosphorous, albumin, alkaline phosphatase, aspartate aminotransferase (AST), alanine aminotransferase (ALT) and creatinine were performed on the day of blood collection. The serum aliquots for the other parameters were stored at −70 °C until the day of the assay. Calcium, phosphorus, alkaline phosphatase, fasting glucose, albumin, AST, ALT and creatinine were determined using an automatic biochemistry analyzer (Wiener lab, CT 600i, Thermo Fisher Scientific, Espoo, Finland). HbA1c and basal insulin levels were measured using an immunoenzymatic assay (Atellica^®^ CH, Siemens Heathineers, Erlangen, Germany). PTH, 25(OH) vitamin D and osteocalcin (Liaison, Diasorin, Saluggia, Italy) levels were measured using chemiluminescence. Leptin and Chemerin (Biovendor, R&D, Karásek, Czech Republic) were determined using ELISA. Serum insulin levels and HOMA-IR were assessed in 23 controls and 10 patients since DS individuals with type 2 diabetes using insulin therapy were not considered for these parameters. Insulin resistance was calculated using the HOMA-IR index: [fasting glucose (mg/dL) × 0.0555 × fasting insulin (µU/mL)/22.5], with normal values < 2.71 [45].

### 4.3. LMNA Mutation Analysis

DNA was extracted from the buccal swabs using standard procedures. The regions of interest were captured using target probes. Next-generation sequencing with Illumina technology was used to investigate a panel of genes, including *LMNA*, *PPARG*, *LIPE*, *PLIN1* and *CIDEC*, among others. The alignment and identification of pathogenic variants were made using bioinformatic protocols, based on the human genome version GRCh38 and processed by ExomeDepth (https://cran.rproject.org/web/packages/ExomeDepth/index.html, accessed on 4 May 2023), a bioinformatic R program that was developed to identify copy number variations (CNVs). The analysis was guided by medical information provided by the laboratory researchers.

### 4.4. Dual-Energy X-ray Absorptiometry

Bone mineral density (BMD) in the lumbar spine (L1–L4), total hip (TH), femoral neck (FN) and 1/3 radius (1/3R) were determined using DXA (GE Healthcare, Lunar Prodigy Advance, Pittsburgh, PA, USA). BMD values were expressed as g/cm² and Z-score since the majority of patients were fertile women and men younger than 50 years. L1–L4 TBS (unitless) to assess bone microarchitecture was measured using the software Insight version 3.0 (Medimaps, Geneva, Switzerland).

### 4.5. ¹H-MR Spectroscopy

#### 4.5.1. Lumbar Spine

Controls and patients underwent spine MRI on a 3.0T system (Philips Medical System, ACHIEVA), using a phased-array coil for the lumbar region. For lumbar spine spectroscopy acquisition, a sagittal T2 weighted (echo time—TE = 120 ms; gap = 4.4mm; echo-train-length = 19; repetition time—TR = 3900 ms; slice thickness = 4 mm, flip angle = 45°) fast spin echo sequence was used as a reference for the placement of a 20 × 20 × 20 mm voxel in the center of the third lumbar vertebral body (L3). The ¹H-MRS proton magnetic resonance spectroscopy was performed using the Point Resolved Spectroscopy (PRESS) technique with 16 acquisitions with water suppression and a flip angle of 90°, as follows: TR: 1600 ms; TE’s: 30/45/60/75/90 ms. Data were processed using LCModel software (Version 6.1, http://www.s-provencher.com/pages/lcmodel.shtml, accessed on 25 September 2021), and the values obtained were used to calculate fat fractions (saturated and unsaturated lipids). The BMAT content and saturated fat fraction in L3 were estimated as previously described by Parreiras-e-Silva et al. [46]. Due to claustrophobia, 5 controls and 4 DS patients did not undergo lumbar spine MRI.

#### 4.5.2. Soleus Muscle

An XL torso coil was positioned over the proximal tibia. Axial T2-weighted fast spin echo acquisition (TR/TE = 400/11 ms; gap = 1.0 mm, slice thickness = 4 mm; field of view = 22 cm) was used as a reference for the spectroscopy voxel placement. A 20 × 20 × 49 mm voxel was placed in the largest area of the soleus muscle to estimate the muscle lipid content also using the PRESS technique with the following parameters: TR/TE = 2400/36 ms, 48 acquisitions with water suppression and 8 acquisitions without water suppression. Intramyocellular (IMCL) and extramyocellular (EMCL) lipids were quantified from the spectra using LCModel software considering the sum of the following peaks: 0.9; 1.3 and 2.1 ppm for IMCL; and 1.1; 1.5 and 2.3 ppm for EMCL. The IMCL and EMCL estimates were automatically scaled to the unsuppressed water peak and later expressed as the IMCL-to-EMCL ratio.

### 4.6. Osteoclastogenesis

Fresh blood samples obtained from 7 controls and 13 DS patients were used to retrieve peripheral blood mononuclear cells (PBMC). Samples were centrifuged and the buffy coat was separated using Histopaque^®^—1.077 g/mL, Sigma-Aldrich^®^ solution, Irvine, UK, according to the manufacturer’s recommendations. Then, the quantification of PBMCs was determined and posteriorly magnetic separation (CD14 MicroBeads, MACS, Miltenyi Biotec©, Teterow, Germany) was used to obtain monocytes from which osteoclasts were derived. These monocytes were cultured in 3 × 3 wells in 96-well plates with α-MEM containing 10% FBS added sequentially in triplets with 25 ng/mL of human macrophage colony-stimulating factor (MCSF) (R&D systems, catalog 216-MCC, Minneapolis, MN, USA), 25 ng/mL of receptor activator of nuclear factor kappa-B ligand (RANKL) (R&D systems, catalog 390-TN, Minneapolis, MN, USA) and 50 ng/mL of chemerin (R&D systems, catalog 2324-CM, Minneapolis, MN, USA) to stimulate osteoclasts. After a period of roughly three weeks, osteoclasts differentiated in the plates and were stained with Giemsa for cell counts. The number of osteoclasts (cells with 3 nuclei or more) was then counted. The wells were photographed, and the pictures were analyzed using ImageJ software version 1.53 to determine the number and area of osteoclasts [47,48].

### 4.7. Statistical Analysis

Statistical analyses were performed, and the figures were generated using R (version 4.1.3) and SAS 9.4. For all analyses, the level of significance was set at *p* < 0.05. Variables in which a logarithmic transformation was used to ascertain normality were expressed as the median, first quartile (25th) and third quartile (75th) (qualitative variables) or as means ± SD (quantitative variables). 

Groups were compared using analysis of covariance (ANCOVA). Normality assumptions were assessed using histograms, dispersion measures and quantile-quantile plots; otherwise, the results were analyzed using logarithmic transformations. When these assumptions were not observed, transformations in the response variable were made for correction. Variables in which a logarithmic transformation was used to ascertain normality were expressed as median and interquartile range. Spearman’s correlation coefficients were calculated to verify associations (ρ) between the variables. 

A multivariate linear regression model was applied, and associations between biochemical, DXA and MRI parameters were assessed via simple and multiple linear regressions before and after adjustment for possible confounding factors (T2D, sex, age and BMI); a beta coefficient (β) was used to estimate the impact of a predictor variable on a dependent variable [49,50].

## Figures and Tables

**Figure 1 ijms-24-13118-f001:**
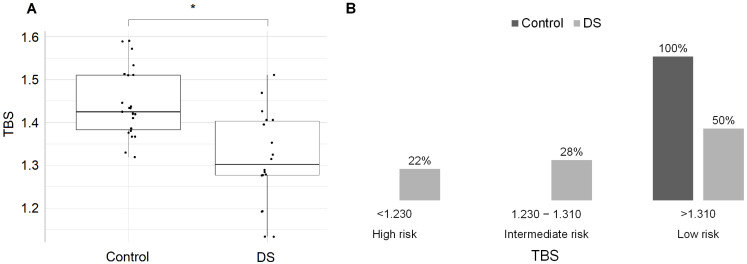
(**A**) Trabecular bone score (TBS) scatter plot of controls (1.440) compared to DS (1.310); * *p* < 0.05. (**B**) Percentual of subjects in each group according to fracture risk, based on TBS thresholds [18]. Abbreviations: DS, Dunnigan syndrome.

**Figure 2 ijms-24-13118-f002:**
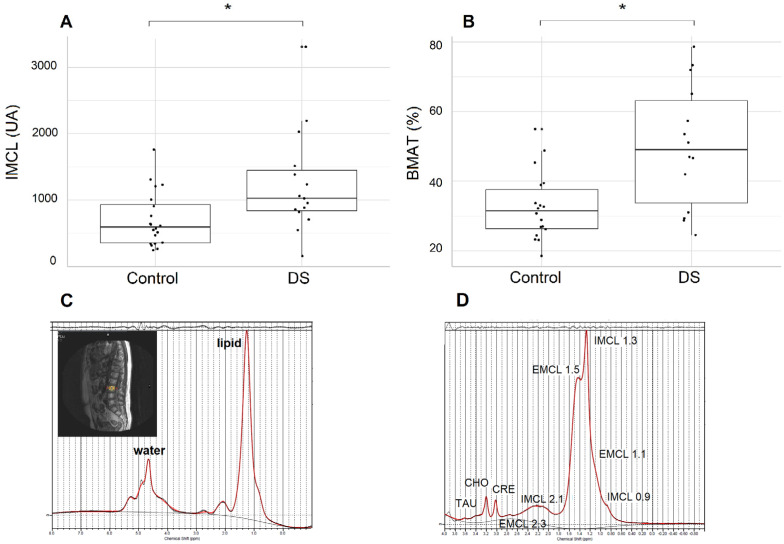
Scatter plot shows that (**A**) intra-myocyte lipids (IMCL) and (**B**) bone marrow adipose tissue (BMAT) are higher in DS patients compared to the control group; (**C**) image of a DS patient generated by ¹H-spectroscopy. The areas under the curve exhibit water and lipid quantifications in the L3 vertebra; (**D**) ¹H-spectroscopy of soleus muscle in the left calf of a DS patient. Areas under the curve display intra- and extra-myocyte lipid quantities. * *p* < 0.05.

**Figure 3 ijms-24-13118-f003:**
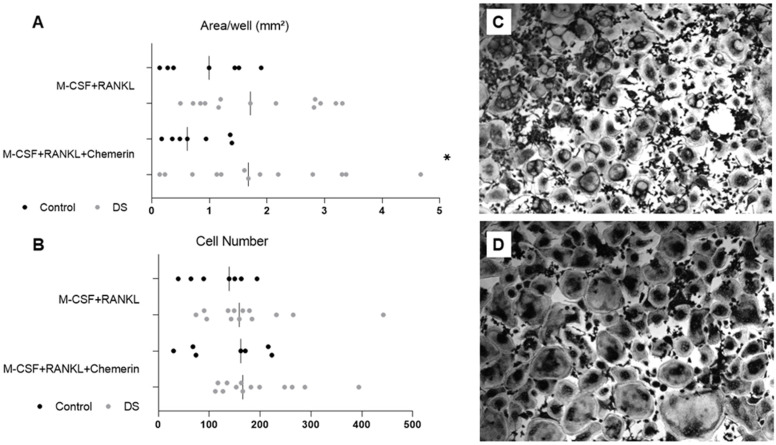
Osteoclast differentiation derived from peripheral blood mononuclear cells from controls and DS patients. (**A**) An area of osteoclasts developed in a culture in the presence of MCSF + RANKL and MCSF + RANKL + chemerin. (**B**) The number of osteoclast cells according to the type of medium added in sequence. Each step of the sequence was cultured in 3 distinct wells for each subject analyzed. The final result is the average sum of controls and patients. (**C**) The appearance of the osteoclast cells developed in a culture supplemented with MCSF + RANKL and (**D**) MCSF + RANKL + chemerin in a DS subject. Osteoclasts must have 3 or more nuclei. * *p* < 0.05. Abbreviations: MCSF, macrophage colony-stimulating factor; RANKL, receptor activator of nuclear factor kappa-B ligand; DS, Dunnigan syndrome.

**Figure 4 ijms-24-13118-f004:**
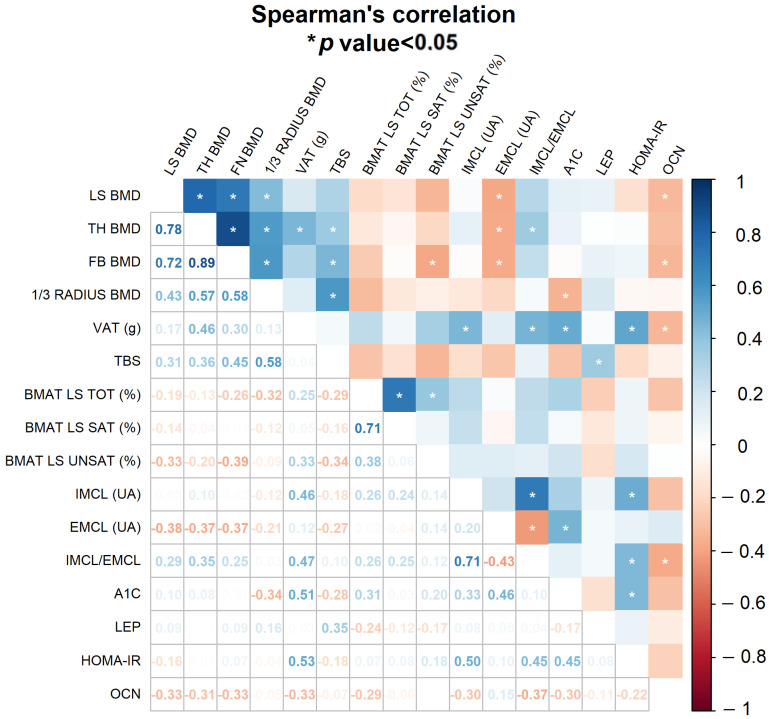
Spearman’s correlation heatmap between bone mineral density, bone microarchitecture, BMAT and biochemical parameters, with color graded scale. * *p* < 0.05. Abbreviations: BMAT, total, saturated and unsaturated bone mineral adipose tissue; EMCL, extramyocellular lipids; LEP, leptin; OCN, osteocalcin; BMD, bone mineral density; LS, lumbar spine; TH, total hip; FN, femoral neck; TBS, trabecular bone score; A1c, glycated hemoglobin; VAT, visceral adipose tissue; HOMA-IR, homeostatic model assessment for insulin resistance; IMCL, intra-myocellular lipids.

**Figure 5 ijms-24-13118-f005:**
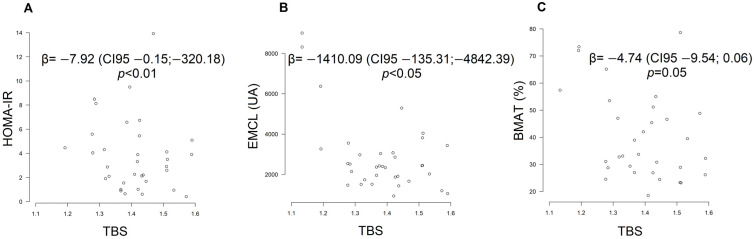
Regression analysis displaying a negative association between HOMA-IR (**A**), EMCL (**B**) and BMAT (**C**) with TBS; (**A**,**B**) shows regression after adjustment for sex, age, and BMI; (**C**) the unadjusted regression analysis. Abbreviations: HOMA-IR, homeostatic model assessment for insulin resistance; TBS, trabecular bone score; EMCL, extra-myocyte lipids; BMAT, total, saturated and unsaturated bone mineral adipose tissue.

**Table 1 ijms-24-13118-t001:** Clinical characteristics of control and Dunnigan syndrome groups.

	Reference Values	Control (*n*= 23)	DS (*n* = 18)	*p*-Value
Female/male		18/5	15/3	
Age, years		38.6 ± 9.6	39.2 ± 11.0	0.76
Height, m		1.67 ± 0.08	1.62 ± 0.09	0.67
Weight, kg		73.9 ± 15.2	70.6 ± 21.2	0.76
BMI, kg/m²	(18–25)	26.3 ± 4.5	26.5 ± 5.2	0.75
Glucose, mg/dL	(70.0–100.0)	91.2 ± 9.7	175.4 ± 107.7	<0.05
HbA1c, %	(4.7–5.6)	5.0 (4.6; 5.6)	7.4 (5.5; 10; 6)	<0.05
Fasting insulin, µIU/mL	(2.0–25.0)	10.7 ± 6.8	26.4 ± 17.0 ^#^	<0.05
HOMA-IR	<2.7	2.1 (1.0; 3.5)	5.6 (4.1; 8.5) ^#^	<0.05
Total cholesterol, mg/dL	<190.0	169.8 ± 30.5	186.4 ± 48.2	0.19
HDLc, mg/dL	>40.0	53.6 ± 13.9	33.1 ± 10.9	<0.05
LDLc, mg/dL	<130.0	97.7 ± 26.5	91.3 ± 32.6	0.50
Triglyceride, mg/dL	<150.0	75.0 (51.0; 116.0)	378.0 (197.0; 612.0)	<0.05
Serum calcium, mg/dL	8.5–10.5	9.0 ± 0.3	10.0 ± 2.4	<0.05
Albumin, g/dL	3.5–5.0	4.4 ± 0.3	4.6 ± 1.4	0.63
Serum phosphorus, mg/dL	2.5–5.5	3.9 ± 0.6	4.5 ± 1.3	0.06
Alkaline phosphatase, U/L	40.0–150.0	58.5 (50.0; 71.8)	101; 5 (75.0; 123.0)	<0.05
Urea, mg/dL	10.0–45.0	29.0 (26.4; 35.8)	28.5 (23.0; 33.0)	0.29
Creatinine, mg/dL	0.6–1.2	0.9 ± 0.2	0.8 ± 0.1	<0.05
AST, U/L	10.0–40.0	21.0 (17.0; 27.0)	21.0 (17.0; 36.0)	0.14
ALT, U/L	5.0–40.0	19.7 (13.6; 28.0)	26.5 (21.0; 56.0)	<0.05
25(OH) vitamin-D, ng/mL	>20.0	25.5 ± 8.3	18.7 ± 8.0	<0.05
PTH, pg/mL	15.0–65.0	69.3 ± 28.6	37.9 ± 15.2	<0.05
Osteocalcin, ng/mL	11.0–46.0	22.6 ± 5.5	17.8 ± 6.1	<0.05
Leptin, ng/mL	6.5–19.1	7.0 (4.1; 32.8)	4.7 (3.8; 12.2)	0.07
Chemerin, ng/mL	0.3–8.0	4.7 ± 0.9	4.8 ± 1.1	0.90

Values are means ± SD or median (Q25; Q75). ^#^ patients using insulin were excluded. Abbreviations: BMI, body mass index; HbA1c, glycated hemoglobin; HOMA-IR, homeostatic model assessment for insulin resistance; HDLc, high-density lipoprotein cholesterol; LDLc, low-density lipoprotein cholesterol; AST, aspartate aminotransferase; ALT, alanine aminotransferase; PTH, parathyroid hormone.

**Table 2 ijms-24-13118-t002:** Densitometric characteristics of the study subjects.

DXA		Control (*n* = 23)	DS (*n* = 18)	*p*-Value
Lumbar spine (L1–L4)	BMD, g/cm²	1.199 ± 0.161	1.180 ± 0.210	0.79
	Z-score. SD	0.2 ± 1.5	0.2 ± 1.7	0.97
Total hip	BMD, g/cm²	1.000 ± 0.146	1.040 ± 0.190	0.50
	Z-score, SD	0.1 ± 1.1	0.4 ± 1.5	0.41
Femoral neck	BMD, g/cm²	1.002 ± 0.132	1.020 ± 0.210	0.56
	Z-score, SD	0.0 ± 1.1	0.4 ± 1.6	0.47
1/3 radius	BMD, g/cm²	0.890 ± 0.090	0.810 ± 0.120	<0.05
	Z-score, SD	−0.1 ± 0.8	−0.9 ± 1.1	<0.05
TBS (L1–L4)		1.440 ± 0.080	1.310 ± 0.110	<0.05

Values are means ± SD. Abbreviations: DS, Dunnigan syndrome; TBS, trabecular bone score.

**Table 3 ijms-24-13118-t003:** Correlation of osteoclast area and number with lumbar spine, total hip and femoral neck bone mineral density (BMD), bone marrow adipose tissue (BMAT) and trabecular bone score (TBS).

	ρ	*p*-Value
Osteoclast area (RANKL) × lumbar spine BMD	0.57	<0.05
Osteoclast area (RANKL + chemerin) × lumbar spine BMD	0.40	0.08
Osteoclast area (RANKL) × total hip BMD	0.55	<0.05
Osteoclast area (RANKL + chemerin) × total hip BMD	0.46	<0.05
Osteoclast area (RANKL) × femoral neck BMD	0.39	0.09
Osteoclast area (RANKL + chemerin) × femoral neck BMD	0.40	0.08
Osteoclast area (RANKL) × BMAT	0.14	0.58
Osteoclast area (RANKL + chemerin) × BMAT	0.11	0.69
Osteoclast area (RANKL) × TBS	0	0.98
Osteoclast area (RANKL + chemerin) × TBS	0.03	0.91
Osteoclast number (RANKL) × BMAT	0.25	0.34
Osteoclast number (RANKL + chemerin) × BMAT	0.14	0.60
Osteoclast number (RANKL) × TBS	0.12	0.63
Osteoclast number (RANKL + chemerin) × TBS	0.10	0.68

**Table 4 ijms-24-13118-t004:** Linear regression results between laboratorial, densitometric and magnetic resonance variables before and after adjustment for sex, age, BMI and diabetes mellitus.

	Unadjusted			Adjusted		
	β Effect	CI	*p*-Value	β Effect	CI	*p*-Value
Glucose × femoral neck BMD	3.96	−12.08:19.99	0.62	12.18	0.71:23.65	<0.05
HOMA-IR × 1/3 radius BMD	−0.43	−0.04:89.5	0.52	−0.90	−0.03:0.57	0.05
Osteocalcin × lumbar spine BMD	−1.01	−2.06:0.04	0.06	−1.04	−1.99:−0.09	<0.05
HOMA-IR × TBS	−2.95	−0.04:818.47	0.25	−7.92	−0.15:−320.18	<0.01
EMCL × TBS	−3577.45	−598.14:−13,308.65	<0.01	−1410.09	−135.31:−4842.39	<0.05
EMCL × lumbar spine BMD	−681.17	−299.67:−706.46	<0.05	−250.57	−68.33:−392.41	<0.05
EMCL × total hip BMD	−774.27	−328.41:−925.64	<0.01	−300.16	−76.91:−576.98	<0.05
EMCL × femoral neck BMD	−681.09	−297.10:−667.18	<0.05	−236.45	−59.36:26.01	0.05

Abbreviations: BMD, bone mineral density; HbA1c, glycated hemoglobin; HOMA-IR, homeostasis model assessment; EMCL, extra-myocyte lipids; IMCL, intra-myocyte lipids.

## Data Availability

The data that support the findings of this study are available from the corresponding author upon reasonable request.

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
