# Peer review of "Refining Evaluation of Bone Mass and Adipose Distribution in Dunnigan Syndrome"

_ijms, 2023, doi:10.3390/ijms241713118_

Round 1

Reviewer 1 Report

Please put the genes in italic font (LMNA);

Please explain further how the pairing of the groups was done. If it was matched for age, sex, weight, and height, why were 23 controls and only 18 DS included?

Please include reference values in Table 1. It is unnecessary to repeat the table results in the text (vitamin D, PTH, osteocalcin, BMD, ...).

Do you have any explanation for the fact that the PTH is lower compared to the control group despite the vitamin D deficiency?

Why only 21% of patients were on pioglitazone? How long have they been using it, and at what doses? Did these patients have different bone assessment results than those who did not use it?

Table 2: Include the number of patients in each group. There was a difference between the groups in the radius, but the Z score was still normal. Is this difference clinically significant? Please include a p-value column for each comparison (in both tables).

To get a better idea of the distribution of TBS values by group, a scatter plot chart would be better than the box plot in Figure 1A. Please consider changing it.

The standard deviations of IMCL and EMCL are high. Do they have a normal distribution? Were parametric statistical methods used for comparison? In Figure 4, Spearman's correlation was used (for variables with non-parametric distribution), but the data were shown as mean and standard deviation (parametric variables) in the tables. Please review this.

At the cortical site (radius), the BMD was lower. The authors state in the discussion that this differs from CGL, however, some data in the literature show similar results. Please include it in the discussion (https://doi.org/10.1016/j.jocd.2016.10.002).

According to the literature, there are around 500 cases of FPLD, considering all types, not only DS. Please correct this information in the discussion.

How was the menstruation of patients with DS? Were amenorrhea or hyperandrogenism frequent? Could this have influenced the results?

Are there any differences (laboratory and imaging tests) between patients with and without a confirmed mutation?

Author Response

We thank the referee for the careful analysis of our manuscript, and for raising a number of points of criticism. We took all these points and made the relevant changes aiming at improving the paper. We present below a detailed set of answers to all the questions.

REVIEWER 1

Please put the genes in italic font (LMNA);

Thank you for your suggestion. All genetic quotations were revised.

Please explain further how the pairing of the groups was done. If it was matched for age, sex, weight, and height, why were 23 controls and only 18 DS included?

We thank you for your suggestion. DS is a rare disease and most studies have few subjects, so we used a convenience sample. The matching is appropriate because, although the sample sizes were different, there weren’t significant differences between groups in mean age, sex, weight or height. This information can be found in table 1, page 3 and in Material and Methods, page 12, lines 382-384.

Please include reference values in Table 1. It is unnecessary to repeat the table results in the text (vitamin D, PTH, osteocalcin, BMD, ...).

Thank you for the contribution. Reference values have been included in table 1 (page 3) and repetitive data in the text has been erased.

Do you have any explanation for the fact that the PTH is lower compared to the control group despite the vitamin D deficiency?

Thank you for the comment. We have added the following information in page 11, lines 322-327:

“The levels of vitamin D may not represent an actual state of deficiency in these patients; there may be an alteration in the sensitivity of the calcium sensor receptor. Moreover, systemic and chronic diseases generally present with low levels of vitamin D, but we have no explanation for the finding of non-elevated PTH in this scenario. Perhaps the decreased fat depots alter the metabolism and bioavailability of vitamin D and its metabolites. This subject deserves more study.”

Why only 21% of patients were on pioglitazone? How long have they been using it, and at what doses? Did these patients have different bone assessment results than those who did not use it?

We would like to thank the reviewer for calling attention to this point. The patients attend a public healthcare clinic at the university hospital, and pioglitazone is not included in the list of free drugs offered to T2DM patients. Therefore, only a few individuals have access to pioglitazone. As such, only three subjects were using pioglitazone at a low dosage (15mg po). In the current version of the manuscript, we have added the following paragraph in page 4, lines 121-130:

“Only three patients, two of whom were from the same family, were using pioglitazone. The small number of individuals using pioglitazone is due to the restricted access to this agent, as it is not included in the list of drugs distributed by the government’s healthcare program. Only one of the three patients used pioglitazone regularly for more than one year, while the others used for a few months. Nevertheless, only one individual from this subgroup has low BMD for their age (LS Z-score -2.1SD; TH Z-score -1.8SD; FN Z-score -1.9SD); the others have normal bone mass. These three patients also had lower body weight (mean=55,3kg) and BMI (mean=22,1) compared to DS. The impact of pioglitazone in FPLD2 bone mass needs further evaluation.

Table 2: Include the number of patients in each group. There was a difference between the groups in the radius, but the Z score was still normal. Is this difference clinically significant? Please include a p-value column for each comparison (in both tables).

We thank you for your consideration. The following information was added in Discussion, page 10, lines 269-273:

“Even though Z-score is normal in both groups, the difference between them is 0.080 g/cm² (9.8%), which is higher than the radius 1/3 least significant change (LSC = 0.033) in the DXA equipment. It is necessary to consider that the samples comprise young adults, meaning that the lower BMD may impact fracture risk in the future.”

To get a better idea of the distribution of TBS values by group, a scatter plot chart would be better than the box plot in Figure 1A. Please consider changing it.

We agree with the reviewer and have changed figure 1A, in page 4, from a box plot to a mixed scatter plot.

The standard deviations of IMCL and EMCL are high. Do they have a normal distribution? Were parametric statistical methods used for comparison? In Figure 4, Spearman's correlation was used (for variables with non-parametric distribution), but the data were shown as mean and standard deviation (parametric variables) in the tables. Please review this.

The authors are in debt to the reviewer for the meticulous analysis. As described in the statistical analysis section, all comparisons were made based on parametric methods. The justification for using Spearman’s correlation in the statistical analysis was based on the presence of outliers that could interfere with the Pearson test. In the current version of the manuscript, we have corrected the information about the test used for the statistical analysis. Below, you can find the modified information:

“Groups were compared using analysis of covariance (ANCOVA). Normality assumptions were assessed using histograms, dispersion measures and quantile-quantile plot; otherwise, the results were analyzed with logarithmic transformations. When these assumptions were not observed, transformations in the response variable were made for correction. Variables in which a logarithmic transformation was used to ascertain normality were expressed as median and interquartile range.”

For instance, IMCL and EMCL were changed to median, interquartile 25 and 75, as suggested by the reviewer. This was added in Statistical Analysis section, page 14, section 4.7, lines 502-507.

At the cortical site (radius), the BMD was lower. The authors state in the discussion that this differs from CGL, however, some data in the literature show similar results. Please include it in the discussion (https://doi.org/10.1016/j.jocd.2016.10.002).

We thank you for your suggestion. Revisions in the text were made to include the suggested article’s information, cited in reference number 25. The following paragraph has been added to the current version of the manuscript in page 11, lines 285-289:

“In a previous study on congenital generalized lipodystrophy, sites with a high content of trabecular bone, such as lumbar spine and ultradistal radius, had high BMDs and Z-scores. On the other hand, the site with the lowest BMD and Z-score was the 1/3 radius. However, in this study there was no control group for comparison.”

According to the literature, there are around 500 cases of FPLD, considering all types, not only DS. Please correct this information in the discussion.

Thank you for your review. The sentence included in the article, in page 12, lines 384-387, is: “Since DS is a rare disease, with approximately 500 reported cases in literature, and FPLD estimated prevalence is of 2.8 cases/million, this study used a convenience sample of FPLD patients who already attended our Diabetes outpatient clinics.”

This information was based on an article by Hussain, I. and Garg, A. (2016), cited in the manuscript as reference 39 (http://dx.doi.org/10.1016/j.ecl.2016.06.012), in which the authors described the prevalence of FPLD2 as the “Most common subtype; more than 500 patients reported” with 3 references to this data, added in the present article in references 40 to 42.

How was the menstruation of patients with DS? Were amenorrhea or hyperandrogenism frequent? Could this have influenced the results?

Thank you for the comment. This information has been added in page 12, lines 388-389:

Fertility, menstrual cycle regularity, and polycystic ovary syndrome were not evaluated in the study.

Are there any differences (laboratory and imaging tests) between patients with and without a confirmed mutation?

We thank the reviewer for calling attention to this point. There was no difference in biochemical and imaging profiles between these subgroups of individuals. This information has been added in page 12, lines 402-409:

“The patients without confirmed mutation were included because they exhibit a clinical and biochemical profile found in FPLD, i.e. metabolic syndrome, presence of DM (glucose mean=179.8 vs 175.4 mg/dL), high triglycerides (mean=477.3 vs 453.7 mg/dL)/low HDLc (mean=34 vs 33 mg/dL) and altered body composition with elevated fat mass ratio index (mean=1.58 vs 1.56); BMD also was comparable (LS 1.125 vs 1.180; TH 1.110 vs 1.040; FN 1.080 vs 1.020; 1/3 radius 0.900 vs 0.810 g/cm²). According to Endocrine Society guidelines, genetic confirmation is not an obligatory criterion for diagnosis of Dunnigan Syndrome.”

Reviewer 2 Report

Mariana Mascarenhas Moreira et al. conducted an interesting study exploring the potential relationship between bone mass and adipose distribution in Dunnigan syndrome. While providing valuable data, the manuscript has multiple concerns that need to be addressed before being considered for publishing.

1.     According to result2.1 and table1, DS patients have significantly lower 25(OH) vitamin D and PTH levels, higher ALP levels, and similar albumin and phosphorus, and calcium levels; please list the calcium values in Table 1 and explain why the DS patients have normal calcium/phosphorus given lower vitamin and PTH levels.

2.     Based on result 2.2 and Figure 3, the osteoclasts in the DS patients seem to have larger cell sizes; please correlate each data point with the corresponding patient’s BMD/TBS/ BMAT values and show the results in Figure 3.  Also, refine Figure 3A/B to present individual points in a scatter plot.

3.     The tables, particularly Table 3, require further refinement to enhance clarity and improve their overall meaning, could consider moving the insignificant data points in the supplement table.

This manuscript needs help with editing from native English editors. 

Author Response

We thank the referee for the careful analysis of our manuscript, and for raising a number of points of criticism. We took all these points and made the relevant changes aiming at improving the paper. We present below a detailed set of answers to all the questions.

REVIEWER 2

  1. According to result 2.1 and table1, DS patients have significantly lower 25(OH) vitamin D and PTH levels, higher ALP levels, and similar albumin and phosphorus, and calcium levels; please list the calcium values in Table 1 and explain why the DS patients have normal calcium/phosphorus given lower vitamin and PTH levels.

The calcium values are already listed in table 1. This information has been added in page 11, lines 322-327:

“The levels of vitamin D may not represent an actual state of deficiency in these patients; there may be an alteration in the sensitivity of the calcium sensor receptor. Moreover, systemic and chronic diseases generally present with low levels of vitamin D, but we have no explanation for the finding of non-elevated PTH in this scenario. Perhaps the decreased fat depots alter the metabolism and bioavailability of vitamin D and its metabolites. This subject deserves more study.”

  1. Based on result 2.2 and Figure 3, the osteoclasts in the DS patients seem to have larger cell sizes; please correlate each data point with the corresponding patient’s BMD/TBS/ BMAT values and show the results in Figure 3. Also, refine Figure 3A/B to present individual points in a scatter plot.

Thank you for your review. The suggested changes in Figure 3A/B were made and the correlation data suggested by the reviewer has been added in Table 3.

The following information has also been included in page 11, lines 354-359:

“A positive correlation between osteoclast area and lumbar spine/total hip BMD that probably reflects the inflammatory systemic status of this patients was seen. Most likely, the peripheral environment has a positive influence on the differentiation of monocytes into osteoclasts in vitro. On the other hand, the results obtained in peripheral monocyte culture do not reflect the osteoclastogenesis in bone microenvironment, as the circulatory levels of osteocalcin suggest a low bone turnover.”

  1. The tables, particularly Table 3, require further refinement to enhance clarity and improve their overall meaning, could consider moving the insignificant data points in the supplement table.

We agree with the reviewer. Table 3 has been changed to Table 4 and adjusted as suggested. In the current version of the manuscript, only results with significant difference remain within Table 4.

Reviewer 3 Report

Moreira et al. aimed to refine the estimation of adipose tissue distribution in individuals diagnosed with Dunnigan Syndrome, verifying quantitatively BMAT and its fractions, muscle adipose tissue (EMCL) and myocyte lipids (IMCL).

Authors have nicely reported their findings, however, I would like to address some minor points.

1. More explanation is needed for results presented in Table 1, especially data with significant differences between control and DS group. I have a question for leptin data, in which SD value is close to the mean of leptin value. Can authors explain this?

2. Authors reported the effect of chemerin on the osteoclast differentiation derived from peripheral blood mononuclear cells from controls and DS patients. However, according to Table 1, chemerin level was not different between control and DS group. Can authors add explanation regarding this? Also, authors may refer to this reference (The chemerin-CMKLR1 axis limits thermogenesis by controlling a beige adipocyte/IL-33/type 2 innate immunity circuit - PubMed (nih.gov)) for more expalanation about chemerin effect on adipocytes.

Minor grammar editing is required

Author Response

We thank the referee for the careful analysis of our manuscript, and for raising a number of points of criticism. We took all these points and made the relevant changes aiming at improving the paper. We present below a detailed set of answers to all the questions raised.

REVIEWER 3

Moreira et al. aimed to refine the estimation of adipose tissue distribution in individuals diagnosed with Dunnigan Syndrome, verifying quantitatively BMAT and its fractions, muscle adipose tissue (EMCL) and myocyte lipids (IMCL).

Authors have nicely reported their findings, however, I would like to address some minor points.

  1. More explanation is needed for results presented in Table 1, especially data with significant differences between control and DS group. I have a question for leptin data, in which SD value is close to the mean of leptin value. Can authors explain this?

Thank you for your review. This information has been added in page 9, lines 259-266 and references 19 and 20:

“The leptin values are related to fat mass depots, and varies according to fat quantities. In the control group, percentual body fat ranged from 22 to 52% and in FPLD ranged from 15 to 36%. This could justify the high variation in results in the serum levels of leptin in both groups. In addition, leptin concentrations in FPLD patients are highly variable and our findings are in line with data published previously by Hegele, R.A. et al, who described a leptin value of 11.1±10.3 ng/mL in FPLD family members (DOI 10.1007/978-3-319-09915-6_18), and Brown, R.J. and Gorden, P., who described values of 5.9±3.0 in FPLD patients (DOI 10.1210/jcem.85.9.6768).”

  1. Authors reported the effect of chemerin on the osteoclast differentiation derived from peripheral blood mononuclear cells from controls and DS patients. However, according to Table 1, chemerin level was not different between control and DS group. Can authors add explanation regarding this? Also, authors may refer to this reference (The chemerin-CMKLR1 axis limits thermogenesis by controlling a beige adipocyte/IL-33/type 2 innate immunity circuit - PubMed (nih.gov)) for more expalanation about chemerin effect on adipocytes.

We thank you for your consideration; we have included the following explanation on page 11, lines 347-354 and the suggested reference (reference 38):

“The inhibition of chemerin and its receptor also showed a positive association with thermogenic beige fat in animal and in vitro models. In the present study, DS patients exhibited elevated BMAT and increased osteoclast area compared to controls when stimulated with chemerin in vitro. However, we did not observe elevated levels of chemerin in individuals with DS. These results most likely reflect a chronic inflammatory status associated with insulin resistance driven by lipodystrophy or a primary resorption abnormality in DS, rather than a direct influence of chemerin.”

Reviewer 4 Report

This cross-sectional study compared bone mass and adipose tissue in subjects with Dunning syndrome and in controls. The data are original and novel. Methodology is described adequately.

1. The number of non-standard abbreviations should be drastically reduced.

2.  Relevance of these subjects for obese individuals should be rephrased. The data in subjects with familial partial lipodystrophy are relevant for subjects with the metabolic syndrome but not for all obese subjects. The common denominator is the presence of insulin resistance and the presence of ectopic lipid deposits and lipotoxicity.

3. Please check whether the format of the references complies with the guidelines of IJMS.

Author Response

We thank the referee for the careful analysis of our manuscript, and for raising a number of points of criticism. We took all these points and made the relevant changes aiming at improving the paper. We present below a detailed set of answers to all the questions raised.

REVIEWER 4

This cross-sectional study compared bone mass and adipose tissue in subjects with Dunning syndrome and in controls. The data are original and novel. Methodology is described adequately.

  1. The number of non-standard abbreviations should be drastically reduced.

We thank you for your review. FG was changed to glucose, OCN was changed to osteocalcin and abbreviations of bone densitometry sites and other terms were reduced whenever possible.

  1. Relevance of these subjects for obese individuals should be rephrased. The data in subjects with familial partial lipodystrophy are relevant for subjects with the metabolic syndrome but not for all obese subjects. The common denominator is the presence of insulin resistance and the presence of ectopic lipid deposits and lipotoxicity.

We thank you for your suggestion and have made the appropriate changes, which can be found in page 10, lines 247-250:

“In the contemporaneous scenario in which metabolic syndrome is on constant rise, FPLD is a rare but important condition that can help delineate the importance of adi-pose tissue to skeletal health, especially considering the impacts of ectopic lipid depos-its and lipotoxicity.”

  1. Please check whether the format of the references complies with the guidelines of IJMS.

Thank you for your review. We have corrected the references according to IJMS guidelines and attached a copy of the patient’s informed consent.